# Genomic epidemiology of *Streptococcus pyogenes* from pharyngeal and skin swabs in Gabon

Sébastien Boutin,[1,2] Benjamin Arnold,[3,4] Abraham Sunday Alabi,[3] Sabine Bélard,[3,5,6] Nicole Toepfner,[7] Dennis Nurjadi[1,8]

**ABSTRACT**    The disease burden of *Streptococcus pyogenes* is particularly high in low- and middle-income countries. However, data on the molecular epidemiology of *S. pyogenes* in such regions, especially sub-Saharan Africa, are scarce. To address this, whole-genome sequencing (WGS) of *S. pyogenes* from Gabon was performed to identify transmission clusters and provide valuable genomic data for public repositories. A total of 76 *S. pyogenes* isolates from 73 patients, collected between September 2012 and January 2013, were characterized by short-read whole-genome sequencing. The predominant emm types were *emm*58.0, *emm*81.2 and *emm*223.0 with 9.2% (7 of 76), 7.9% (6 of 76), and 6.6% (5 of 76), respectively. Single-nucleotide polymorphism analysis revealed 16 putative transmission clusters. Four of these were household transmissions. Four antimicrobial genes (*lmrP*, *tetM*, *tetL*, and *thfT*) were found in the *S. pyogenes* isolates from this study. All strains carried *lmrP*. Of the 76 isolates, 64 (84.2%) carried at least one tetracycline resistance gene (*tetM* or *tetL*). Comparisons with other publicly available African genomic data revealed a significant correlation between geographical location and genetic diversity of *S. pyogenes*, with Gabonese strains showing similarities to those from Kenya and certain Oceanian regions. Our study showed that transmission of *S. pyogenes* can occur at the community/household level and that high-resolution molecular typing is needed to monitor changes in circulating clones and to detect community outbreaks. Advocacy for the adoption of WGS for comprehensive molecular characterization of *S. pyogenes* and data sharing through public repositories should be encouraged to understand the molecular epidemiology and evolutionary trajectory of *S. pyogenes* in sub-Saharan Africa.

**IMPORTANCE**    The study conducted in Gabon underscores the critical importance of addressing the limited knowledge of the molecular epidemiology of *Streptococcus pyogenes* in low- and middle-income countries, particularly sub-Saharan Africa. Our molecular analysis identified predominant *emm* types and unveiled 16 putative transmission clusters, four involving household transmissions. Furthermore, the study revealed a correlation between geographical location and genetic diversity, emphasizing the necessity for a comprehensive understanding of the molecular epidemiology and evolutionary trajectory of *S. pyogenes* in various regions. The call for advocacy in adopting whole-genome sequencing for molecular characterization and data sharing through public repositories is crucial for advancing our knowledge and implementing effective strategies to combat the spread of *S. pyogenes* in sub-Saharan Africa.

**KEYWORDS**    group A streptococcus, *Streptococcus pyogenes*, Africa, molecular epidemiology, whole-genome sequencing

*S*treptococcus pyogenes, also known as group A *Streptococcus* (GAS), is a Gram-positive bacterium that can cause a wide range of infections in humans. The clinical presentation of *S. pyogenes* infections can range from mild conditions like pharyngitis

Address correspondence to Dennis Nurjadi, dennis.nurjadi@uni-luebeck.de.

Sébastien Boutin and Benjamin Arnold contributed equally to this article. Order of authors was determined by increasing seniority.

Nicole Toepfner and Dennis Nurjadi contributed equally to this article.

The authors declare no conflict of interest.

See the funding table on p. 8.

and impetigo contagiosa to more severe and life-threatening infections, such as necrotizing fasciitis and septic shock. *S. pyogenes* infections are most common in children and adolescents. In some cases, *S. pyogenes* infections can lead to severe post-streptococcal complications or immunogenic sequelae, such as acute rheumatic fever, post-streptococcal glomerulonephritis, and even toxic shock syndrome (1).

The burden of diseases and mortality due to *S. pyogenes* infections are exceptionally high in low- and middle-income country (LMIC) settings. Data on molecular epidemiology can help to study the evolutionary dynamics of bacterial populations and are important for developing effective prevention and treatment strategies to combat *S. pyogenes* infections in resource-limited settings, such as in sub-Saharan Africa. However, there is an apparent data disparity in the molecular epidemiology of *S. pyogenes* in LMIC compared to high-income countries (HICs) (2). Upon searching the public repositories, only limited sequence data of *S. pyogenes* from sub-Saharan Africa are available on public repositories (3).

Based on these observations, we performed whole-genome sequencing (WGS) to characterize *S. pyogenes* collected from a previous study in Gabon, Africa (4, 5). Since *S. pyogenes* infections are mostly community-acquired via direct contact or via droplets, the main objective was to identify potential transmission clusters, which may provide insights into the transmission dynamics and spread of specific clonal lineages in this community. Lastly and most importantly, by performing WGS, we aim to contribute valuable genomic data on *S. pyogenes* from Central Africa to the public repository.

## MATERIALS AND METHODS

### Study population and microbiology

Bacterial isolates and clinical data were collected in a previous study performed at the Centre de Recherches Médicales de Lambaréné (CERMEL) in Gabon, Central Africa, as previously described (4). The study was performed between September 2012 and January 2013. Participants were recruited within the province of Moyen-Ogooué in Gabon. Recruitment, interviews, and sampling took place at the participants' homes, which were consecutively approached by researchers within one rainy season. Inclusion criteria were (i) provision of written informed consent and (ii) residence within the province of Moyen-Ogooué. For this study, the species identification and confirmation from the frozen bacterial isolates were performed using mass spectrometry (Matrix-Assisted Laser Desorption Ionization Time-of-Flight Mass Spectrometry [MALDI-TOF MS]; Bruker, Germany). Only non-duplicate isolates (one isolate per patient per site) confirmed as *S. pyogenes* were selected for WGS ($n = 76$). Antibiotic susceptibility testing was performed by disk diffusion for penicillin, cefotaxime, clindamycin, erythromycin, azithromycin, and tetracycline, as described previously (5). The antibiotic susceptibility was interpreted according to the European Committee on Antimicrobial Susceptibility Testing (EUCAST) clinical breakpoints of the respective year.

### Definition of infection and asymptomatic carriage

Children and adults living in rural and urban areas of the province of Moyen-Ogooué were recruited through spontaneous home visits; i.e., samples were collected from infected and asymptomatic carriers. The McIsaac score and the presence of rhinitis, conjunctivitis, and skin infection were assessed from the demographic and clinical data collected. A throat swab and, in the case of skin infection, an additional skin swab were taken from each participant. Tonsillopharyngitis was defined as isolation of *S. pyogenes* from the throat swab of study participants older than 3 years of age who presented with sore throat on the day of sample collection and had a McIsaac score ≥3, as previously described (5). Asymptomatic carrier status (colonization) was defined as isolation of *S. pyogenes* from the throat swab but absence of any of the criteria defining GAS tonsillopharyngitis. Skin infection caused by *S. pyogenes* was defined as a pyogenic skin lesion and isolation of *S. pyogenes* from that site.

## Whole-genome sequencing and data analysis

For each *S. pyogenes* isolate, the DNA extraction, library preparation, sequencing on a MiSeq Illumina platform (short-read sequencing, 2 × 300 bp), and post-sequencing procedure were performed as previously described. Briefly, raw sequences were controlled for quality using fastp (v.0.23.2 with parameters −*q* = 30 and −*l* = 45) and assembled with SPAdes (v.3.15.5) (with the option—careful and—only-assembler). Draft genomes were curated by removing contigs with a length of <500 bp and/or coverage of <10× . The quality of the final draft genome was quality-controlled using Quast (v.5.0.2). The complete draft genomes were processed through available databases using Abricate (https://github.com/tseemann/abricate) to identify antimicrobial resistance (National Center for Biotechnology Information, CARD, ARG-ANNOT, ResFinder, and MEGARES databases) and virulence factors (Virulence Factor DataBase, [VFDB]). The species identification of each draft genome was done using mash (sub-command screen) by screening each draft genome to a database composed of a representative genome of each species present in the Microbial Genomes resource (https://www.ncbi.nlm.nih.gov/genome/microbes/), and the *emm* types and *emm* clusters were determined using emm-typer (https://github.com/MDU-PHL/emmtyper). A phylogenetic comparison of the *S. pyogenes* strains from Gabon to publicly available *S. pyogenes* genomes was performed using core-genome analysis. In short, 2,112 public genomes from *S. pyogenes* were found with a geographical location from the Refseq bacteria database. Each genome was annotated using Prokka (v.1.14.5), and the core genome was calculated using Roary (v.3.13.0). Genes present in more than 90% of the population were kept in the core-genome alignment and used to build a phylogenetic tree using Raxml. Furthermore, a more granular phylogeny was performed on the *S. pyogenes* strains from this study, where each draft genome was aligned to the representative *S. pyogenes* genome reference using SKA. The alignment was then analyzed with Gubbins (v.3.2.1) to define single-nucleotide polymorphism (SNP) distance. A threshold of 18 SNPs was considered as the upper limit for transmission events based on the minimum spanning tree. This threshold was mostly concordant with the clustering observed using average nucleotide identity with a threshold of 99.99% as described previously (6) (Fig. S1).

## RESULTS

### Molecular characteristics of gabonese *S. pyogenes* strains

A total of 76 *S. pyogenes* isolates were sequenced, originating from 73 patients. Of these, 62 isolates were obtained from throat swabs, while 14 were from skin swabs. Our study involved 12 household clusters: 1 with four members (1 of 12, 8.3%), 2 with three members each (2 of 12, 16.7%), and 9 with two members each (9 of 12, 75%). The predominant emm types were *emm*58.0, *emm*81.2, and *emm*223.0 with 9.2% (7 of 76), 7.9% (6 of 76), and 6.6% (5 of 76), respectively. Previously, these isolates underwent *emm*-typing using PCR and Sanger sequencing (5). For 64 of the 76 isolates, WGS-derived *emm* typing and conventional *emm*-typing method delivered concordant results (Fig. S2). Minor deviations due to sub-typing were observed in four isolates with *emm*65 by PCR and emm65.1 by WGS, and four isolates with *emm*209, which was assigned to *emm*209.1 by WGS. Mismatch in two cases, isolate G0465T (*emm*171.1 by PCR and *emm*166.4 by WGS) and G0871T (*emm*1750.0 by PCR and *emm*166.4 by WGS) was most likely due to homologous genes, *enn* and *mrp* (7). Based on homologous genes, the WGS-derived *emm* typing resulted in *emm*171.1 and *emm*1750, respectively. Two major mismatches were observed for two isolates, G078S (*emm*55 by PCR and *emm*122 by WGS) and G0874T (*emm*4.2 by PCR and *emm*4.5 by WGS). A potential explanation may be the presence of multiple clones in the sample. Two *emm* types (*emm*63.3 and *emm*85.0) were found only on skin swabs (*n* = 1 for each *emm* type). Ranked by prevalence, the following *emm* types were found only in the skin: emm58.0, emm81.2, emm92.0, emm209.1, emm22.0, emm11.0, emm12.0, emm166.4, emm181.0, emm4.5, emm93.0, emm119.2, emm132.0, emm18.21, emm224.0, emm56.0, emm63.0, and emm84.1. There

were no statistically significant associations between *emm* types and the site of sample collection (Fisher test, *P* value = 0.1).

Analyzing SNPs across all isolates, we identified 16 potential transmission clusters (≤18 SNPs). Most clusters comprised fewer than three individuals. Notable SNP_clusters were 4 (*n* = 4), 11 (*n* = 6), 15 (*n* = 7), and 16 (*n* = 5), correlating with *emm* clusters E2 (*emm*92.0), E6 (*emm*81.2), E3 (*emm*58.0), and D4 (*emm*223.0), respectively (Fig. 1). Three individuals presented genetically identical clones from both throat and skin (SNP_clusters 7, 12, and 13). The geographical location (village) showed no significant association with the SNP clusters (SNP_clusters 1, 3, 7, 10, 11, 12, and 15), suggesting potential regional circulation of clones in Gabon.

## Household transmissions

Isolates collected from the same household are grouped into household_clusters (*n* = 12, Fig. 2). In four of the 12 household_clusters (household_clusters 2, 8, 10, and 12), genomic analysis suggested intra-household transmission. Three isolates were isolated from members of household_cluster 2, with a total number of 25 residents, of which 2 (ST404/*emm*11.0) were genetically closely related; the third isolate belonged to *emm*65.7 and was therefore unlikely to be related to the other two isolates. Both *emm*11.0 isolates were isolated from the throat swabs of two 4-year-old girls who were asymptomatic carriers. One of the girls had reported a history of skin infection (impetigo) in the past. However, no lesions were present at the time of sampling. Similarly, household_cluster 10 (total number of residents, *n* = 10) had two out of three isolates with ST771/*emm*4.5 and one isolate with *emm*166.4. The two *emm*4.5 isolates were probably transmitted within the household and were isolated from an asymptomatic carrier (5-year-old boy) and from a 22-year-old man with symptoms of tonsillopharyngitis (sore throat).

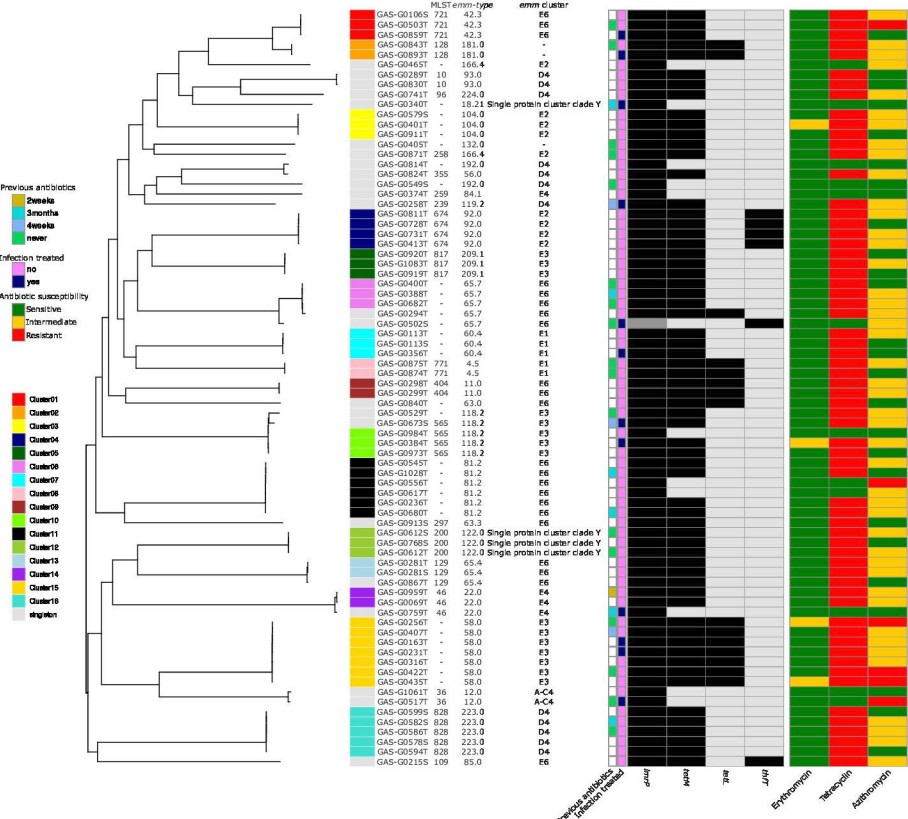

**FIG 1** SNP-based phylogeny of *S. pyogenes* in Gabon. A cluster was defined as SNP ≤18. The presence of antibiotic resistance genes (ARGs) are indicated by black boxes and the absence of ARGs are indicated by light grey boxes in the heatmap.

Transmission was also highly likely in household_clusters 8 and 12, with both clusters (*n* = 2 each) corresponding to ST674/*emm*92.0 and ST817/*emm*209.1, respectively (Fig. 2). Both children (male, ages 5 and 9) in household_cluster 8 (total number of residents, *n* = 12) were asymptomatic carriers. In household_cluster 12 (total number of residents, *n* = 8), the two isolates were collected from the throat swabs of asymptomatic carriers (male, age 9, and female, age 13). The transmission clusters consisted of asymptomatic carriers and infected individuals, but almost all household transmission clusters involved individuals with previous *S. pyogenes* infections. Notably, only one person in household cluster 10 had no history of *S. pyogenes* infection.

## Antibiotic resistance and virulence genes

Four antimicrobial genes (*lmrP*, *tetM*, *tetL*, and *thfT*) were found in the *S. pyogenes* isolates in Gabon. All the strains (76 of 76, 100%) carried an *lmrP* gene, a broad-spectrum drug efflux gene. The tetracycline resistance genes, *TetM* and *tetL*, were also highly prevalent [*n* = 64 (84.2%) and *n* = 15 (19.7%), respectively]. Finally, the gene *thfT*, an energy-coupling factor transporter S component gene conferring sulfonamide resistance, was identified in six strains (7.9%). We did not observe any associations between the AMR prevalence and the usage of antibiotics in the past or previous treatment of infection (Fig. 1). Phenotypic AST was performed for penicillin, cefotaxime, clindamycin, erythromycin, azithromycin, and tetracycline. All isolates were susceptible to penicillin, cefotaxime, and clindamycin. All 64 isolates harboring either *tetM* or *tetL* were phenotypically resistant to tetracycline. Six of 76 (7.9%) isolates were phenotypically resistant to azithromycin. The comparison of phenotypic and genotypic susceptibility to erythromycin, azithromycin, and tetracyclin is displayed in Fig. 1. We did not detect any multidrug-resistant (MDR) (definition: resistant to ≥3 antibiotic drug classes) isolates in our study.

We evaluated the impact of the presence/absence of virulence factors in the genome on the likelihood of infection. We analyzed 65 virulence factors within the genomes of our study population. Thirty-one genes belonged to the core virulome of the population and 34 genes were accessory. None of the genes showed a significant association with the presence of an ongoing infection on the skin nor the throat (Fig. S3).

All strains harbored one or more genes encoding the streptococcal pyrogenic exotoxin (*spe*), namely, 100% (76 of 76) harbored *speB*; 23.7% (18 of 76) harbored

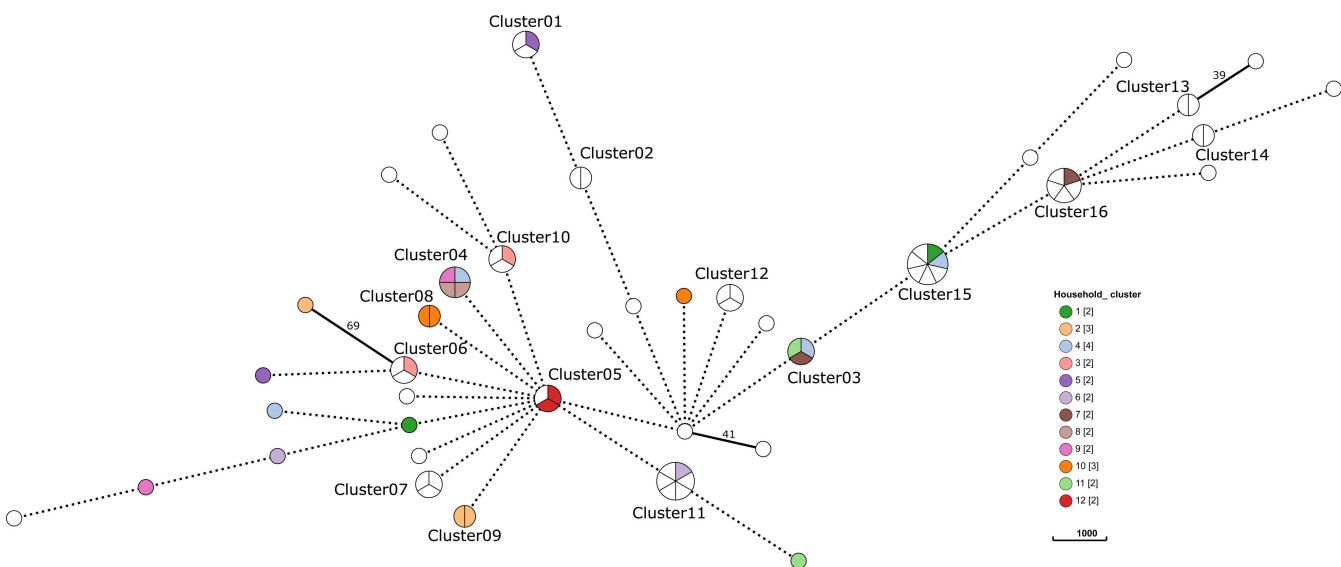

**FIG 2** Association between household and clonal clusters. The minimum spanning tree is based on the SNP distribution among isolates. The branches longer than 100 SNPs are shortened to improve visualization (dotted lines). Branches shorter than 18 SNPs are merged to represent the clonal clusters, and the dots are colored by household cluster.

*speA*; 30.3% (23 of 76) *speC*; 96.0% (73 of 76) *speG* with one strain with a truncated gene (84.68% of the reference *speG* gene); 19.7% (15 of 76) *speH*; 14.5% (11 of 76) *speI*; 30.3% (23 of 76) *speJ*; 9.2% (7 of 76) *speK*; 9.2% (7 of 76) *speL*; and 9.2% (7 of 76) *speM*. Furthermore, 68 of 76 (89.5%) harbored the gene encoding streptococcal mitogenic exotoxin Z (*smeZ*), of which 5 isolates harbored a disrupted *smeZ* gene (nucleotide identity ≤99.6% to the reference/intact *smeZ* gene). We identified several strains harboring more than five superantigen genes (12 of 76, 15.8%). Two isolates belonging to ST10 with the *emm* type 93.0 harbored the most superantigen genes (eight *spe* gene variants and *smeZ*), namely, *speB*, *speC*, *speG*, *speH*, *speI*, *speJ*, *speL*, and *speM*. The patients originated from different neigborhoods (Makouké and Adouma), suggesting that this clone may be circulating locally. Other *emm* types harboring multiple *spe* genes were ST128/*emm*181.0 and ST355/*emm*56.0 (*n* = 7 genes) (Fig. 1; Fig. S4).

## Comparison to other publicly available genomic data from Africa

We observed a significant correlation between geographical location and genetic diversity of *S. pyogenes* isolates in the Gabonese population (permutational multivariate analysis of variance: $R^2$=0.05, *P* value <0.001; Fig. 3). The *S. pyogenes* African strains from Gabon and Kenya tend to cluster together within the phylogenetic tree. This result is in line with the analysis of the prevalence of the *emm* types, the strain from he Gabon cluster with the population of Kenya and Oceania (Fiji, Australia, and New Zealand). However, the prevalence of *emm*81.2, *emm*118.2, *emm*65.7, and *emm*92.0 was higher in the Gabonese population (Fig. S3). Based on the prevalence of *emm* clusters, the Gabonese population clustered with the Kenyan and Oceanian populations again with the addition of the UK population. The highest prevalence in the population of Gabon is observed for clusters E2, E3, E6, and D4, which is concordant with the major clusters identified in this study (Fig. S4).

## Theoretical vaccination coverage in Gabon

We calculated the theoretical coverage of the 30-valent M protein-based *Streptococcus pyogenes* vaccine (8, 9) in our study population. Of the 30 different *emm* types found in our study, 9 (30%) would be covered by this vaccine, while an additional 4 included *emm* types showing cross-opsionization as previously described (8). Overall, in the population, 31 patients (41%) would have benefited from the vaccine and 12 other patients would have been protected, based on the cross-opsionic effect (57% overall).

## DISCUSSION

This study offers insights into the genomic epidemiology of *S. pyogenes* isolated from Gabon, Central Africa. The prevalent *emm* types in Europe and North America, such as *emm*1, *emm*28, *emm*89, *emm*3, *emm*12, *emm*4, and *emm*6, were less common among our Gabonese isolates, which is concordant with other published studies (10). Despite the limited genomic data available for *S. pyogenes* strains from the African continent, our isolates, when put in context with publicly available worldwide genomic data for *S. pyogenes*, clustered with Kenyan and Oceanian isolates but not with other isolates from HIC settings (Fig. 3). The reasons for the differential prevalence of certain clones across regions remain unclear. However, this observation is of high interest as also the disease burden varies between LMIC and HIC. Plausible explanations for the different prevalences of certain clones in LMIC and HIC could stem from disparate data availability, host or bacterial genetic factors influencing colonization and infection, and augmented community and household transmissions in distinct populations. Several studies have suggested the influence of host-microbe interactions in facilitating the pharyngeal persistence of *S. pyogenes* and other bacteria inhabiting the nasopharyngeal compartment (11). Besides comphrehensive genome studies on the pathogen side of infection, further efforts are needed to investigate the role of host immunity and

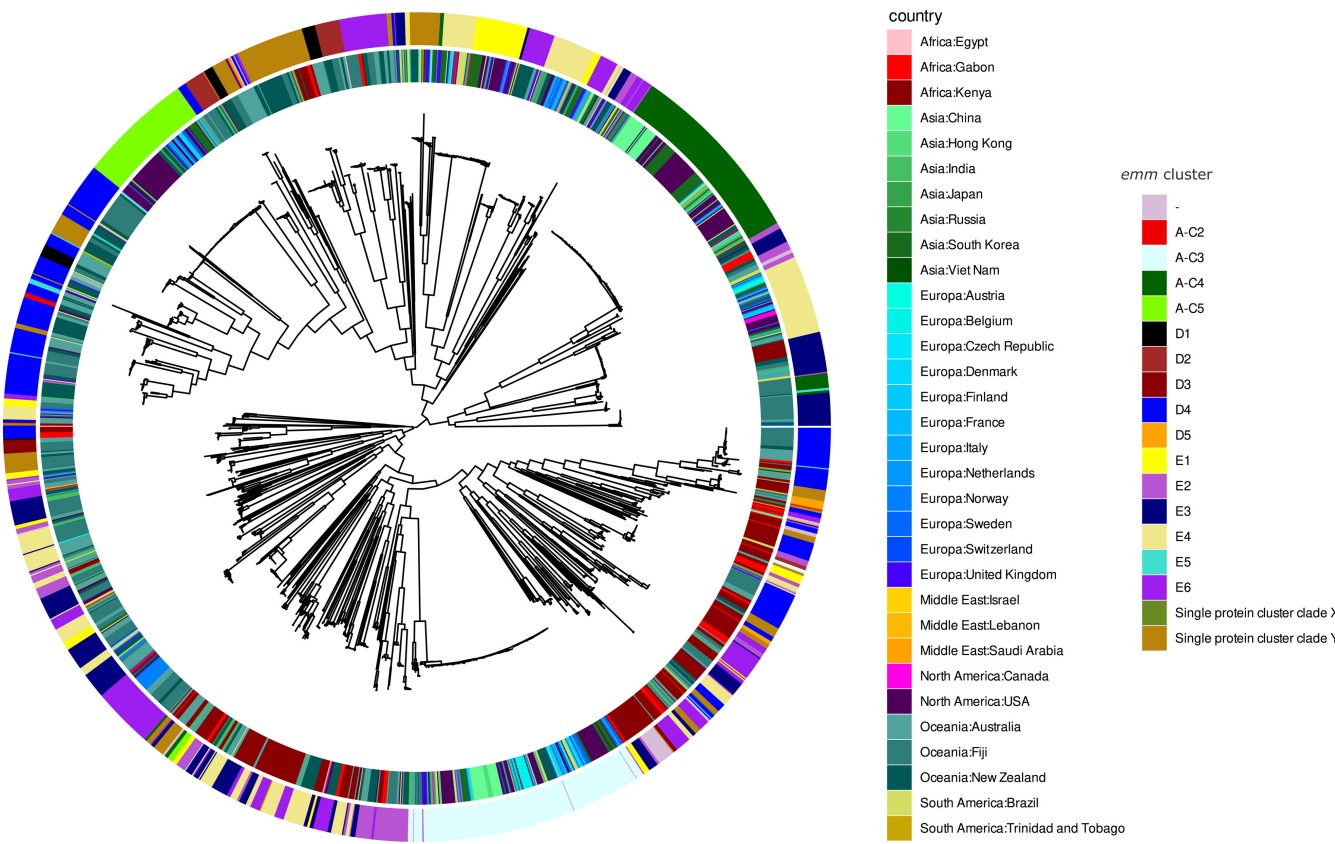

**FIG 3** Phylogeny of *S. pyogenes* worldwide. The internal circle represents the country of isolation, and the external circle represents the *emm* cluster. Data from this study are shown in red (Africa:Gabon), compared with WGS data from public respositories with geographical information.

microbial interactions as drivers of clone evolution in different communities of LMIC and HIC.

Among the circulating *S. pyogenes* clones, some are considered to be virulent or "high-risk" clones, such as the globally circulating M1 (*emm* type 1) clone with enhanced toxigenic properties (10). Although we did not find any *emm*1 clones in our study, we identified several *emm* types, which were associated with the abundance of *spe* genes. The most prominent example is the ST10 *emm*93.0 clones, which harbored eight *spe* genes and a *smeZ* gene. The *emm*93.0 type has been rarely reported. To the best of our knowledge, the *emm93.0* type has only been reported in Spain (12), Kenya (13), Israel (14), and New Caledonia (15). The emergence of an invasive and MDR *emm*93.0 clone harboring a genomic island with the resistance genes *lsa(E)*, *lnu(B)*, *ant (6)-Ia*, and *aph(3′)-III* has been reported in Israel. Compared to the molecular characteristics of the Israeli strains, the Gabonese *emm*93.0 was not multidrug resistant and harbored only *tetM*. It is possible to assume that other variants of the MDR *emm*93.0 are circulating globally, and the evolutionary trajectory of this clone of interest and its contribution to inter-species transfer of antibiotic resistance genes (ARGs) and global ARG spread should be monitored closely.

M protein encoded by the *emm* gene is a key virulence factor of GAS and plays an important role in the pathogenesis of GAS infections for immune evasion (16, 17). Due to the high variability of this gene, *emm* typing has become the most common tool used today to characterize the genetic diversity among *S. pyogenes* isolates (18, 19). Our data suggest that the MLST scheme for *S. pyogenes* displayed a similar resolution to the *emm* types, supporting the use of *emm* typing nomenclature for describing the genomic epidemiology of *S. pyogenes*. Meanwhile, WGS has become an increasingly popular, highly discriminative tool to investigate transmissions and confirm close

relatedness among isolates (20). In this study, we could demonstrate that the *emm* types were concordant within the SNP cluster based on the ≤18 SNP cut-off value for close genetic relatedness. Although *emm* typing was less discriminatory than WGS in defining genetic relatedness (21, 22), *emm* typing can be used for surveillance to identify putative transmission and clonal spread. Nonetheless, we believe that WGS data for *S. pyogenes* are very valuable for investigating at a fine resolution the molecular epidemiology in the community to monitor local and regional emergence, relapsing versus independent *S. pyogenes* infections causing recurrent episodes in certain individuals and household clusters, as well as the global spread of high-risk clones.

Our study does come with certain limitations that need to be acknowledged. While the integration of clinical data with WGS holds the potential to become a robust tool for pinpointing pathogenic determinants linked to specific clinical trajectories, outcomes, or infection severities, it is important to note that our investigation has its constraints. Unfortunately, considering the small study population and mostly carrier isolates without clinical data of infection sub-types, we could not correlate the genome-wide association analysis with different clinical phenotypes of *S. pyogenes* infections. In addition, isolates were collected between 2012 and 2013, which may not reflect the current epidemiology of circulating strains in Gabon. Nonetheless, we believe that our WGS data are valuable in bridging the data gap for *S. pyogenes* molecular epidemiology between HICs and LMICs. Advocating for the adoption of WGS for the comprehensive molecular characterization of *S. pyogenes*, along with the earnest sharing of data, particularly from LMICs, is of paramount importance and should be encouraged to understand the molecular epidemiology and evolutionary trajectory of *S. pyogenes*.

## ACKNOWLEDGMENTS

Project cooperation was established in the Network Young Infection Medicine.

The sample collection and recruitment were supported by the German Society of Pediatric Infectious Disease (B.A. and N.T.)

## AUTHOR AFFILIATIONS

[1]Department of Infectious Diseases and Microbiology, University of Lübeck and University Medical Center of Schleswig-Holstein Campus Lübeck, Lübeck, Germany

[2]Airway Research Center North (ARCN), German Center for Lung Research (DZL), Lübeck, Germany

[3]Centre de Recherches Médicales de Lambaréné (CERMEL), Lambaréné, Gabon

[4]Department of Infectious Disease and Tropical Medicine, St. Georg Hospital, Leipzig, Germany

[5]Institute of Tropical Medicine, University of Tübingen, Tubingen, Germany

[6]German Center for Infection Research (DZIF), Partner Site Tübingen, Tuebingen, Germany

[7]Department of Pediatrics, Faculty of Medicine and University Hospital Carl Gustav Carus, Technische Universität Dresden, Dresden, Germany

[8]German Center for Infection Research (DZIF), Partner Site Hamburg-Lübeck-Borstel-Riems, Lübeck, Germany

## AUTHOR ORCIDs

Sébastien Boutin http://orcid.org/0000-0002-0499-2460
Nicole Toepfner http://orcid.org/0000-0002-9693-4419
Dennis Nurjadi http://orcid.org/0000-0002-1278-5939

## FUNDING

| Funder | Grant(s) | Author(s) |
|---|---|---|
| German Society of Pediatric Infectious Disease (DGPI) | | Benjamin Arnold |

| Funder | Grant(s) | Author(s) |
|---|---|---|
| | | Nicole Toepfner |

## DATA AVAILABILITY

*S. pyogenes* sequence data were uploaded to the National Center for Biotechnology Information GenBank repository under the bioproject number PRJNA1029911. The accession numbers, sequencing statistics, and quality parameters are summarized in Table S1.

## ETHICS APPROVAL

The study was approved by the CERMEL Institutional Review Board. Written informed consent was obtained for all participants before inclusion. For minors, a parent or a legal guardian provided written consent.

## ADDITIONAL FILES

The following material is available online.

### Supplemental Material

**Supplemental material (Spectrum04265-23-s0001.docx).** Fig. S1 to S5.
**Table S1 (Spectrum04265-23-s0002.xlsx).** Quality control, sequencing statistics, Inc types, virulence genes, and antibiotic resistance genes.

### Open Peer Review

**PEER REVIEW HISTORY (review-history.pdf).** An accounting of the reviewer comments and feedback.

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
