## [Reviewer comments · Microbiology Spectrum]

Microbiology Spectrum

Genomic epidemiology of *Streptococcus pyogenes* from pharyngeal and skin swabs in Gabon

Sébastien Boutin, Benjamin Arnold, Abraham Alabi, Sabine Bélard, Nicole Toepfner, and Dennis Nurjadi

Corresponding Author(s): Dennis Nurjadi, Universitat zu Lubeck

Review Timeline:

Submission Date:	December 22, 2023
Editorial Decision:	March 13, 2024
Revision Received:	April 9, 2024
Accepted:	April 19, 2024

Editor: Deena Altman

Reviewer(s): The reviewers have opted to remain anonymous.

Transaction Report:

DOI: <https://doi.org/10.1128/spectrum.04265-23>

Re: Spectrum04265-23 (Genomic epidemiology of *Streptococcus pyogenes* from pharyngeal and skin swabs in Gabon)

Dear Prof. Dennis Nurjadi:

Thank you for the privilege of reviewing your work. Below you will find my comments, instructions from the Spectrum editorial office, and the reviewer comments.

Revision Guidelines

Sincerely,
Deena Altman
Editor
Microbiology Spectrum

Reviewer #1 (Public repository details (Required)):

NCBI for sequence data

Reviewer #1 (Comments for the Author):

In this manuscript, the authors describe population genomic features of group A streptococcus strains recovered from Gabon. Since there are limited studies performed in Africa, this provides new information to the field.

A few suggestions for improvement.

1. Please clarify if the strains were from infected patients or asymptomatic carriers. In particular, please provide more detail on the strains recovered from multiple members of the same household.
2. From the sequence data, it could be informative to also perform analysis of mobile elements such as phage, ICE, etc.
3. Line 84 - what is meant by a smear infection?
4. Line 148 - how was the threshold of 18 SNPs determined?
5. Line 157 - I'm not convinced that transmission equates to virulence. What about disease severity? Related to comment 1, it is not clear if the intrahousehold strains are related to infections or carriage.
6. An additional limitation to discuss is that the strains were recovered in 2012, so the data may not be representative of currently circulating strains.

Reviewer #2 (Comments for the Author):

The study of Boutin S et al. covers an important topic that is the molecular epidemiology of *S. pyogenes* in low and middle-income countries, partially filling a gap in the knowledge of the diffusion of this pathogen in Sub-Saharan Africa. *Streptococcus pyogenes* is an important health concern, being responsible for over 500,000 deaths per year, and is the principal etiological agent of children pharyngitis with many severe sequelae. However, I am some concerns about it that need to be solved before considering for publication, besides the limitations mentioned by the Authors (small study population and absence of clinical data linked to the isolated strains).

Major issues:

1. Have you found some difference in emm types identified in throat and skin swabs?
2. In Antibiotic resistance and virulence genes you affirm:
We do not observe any associations between the AMR prevalence and the usage of antibiotics in the past or previous treatment of infection (Figure 1).
But how much is the AMR prevalence and which antibiotics were used?
It is difficult extrapolate from the Figure 1 such information.
3. Have you performed antibiotic susceptibilities for each strain?
4. Are they multi drug resistant?
5. Have you estimated the possible vaccine coverage with the identified emm types?

Minor issues:

To a better understanding of your results, it is advisable to have some tables for antibiotic and virulence genes

Point-by-point response to the reviewers' comments.

>>Thank you for the opportunity to revise and improve our manuscript. We hope to have addressed the reviewers' comments adequately.

Reviewer #1 (Comments for the Author):

In this manuscript, the authors describe population genomic features of group A streptococcus strains recovered from Gabon. Since there are limited studies performed in Africa, this provides new information to the field.

A few suggestions for improvement.

1. Please clarify if the strains were from infected patients or asymptomatic carriers. In particular, please provide more detail on the strains recovered from multiple members of the same household.

>> **Response:** All strains analyzed in this study were collected during spontaneous household visits in the community, covering both infected patients and asymptomatic carriers. Tonsillitis was defined by a clinical McIsaac score > 3 and GAS isolation from the tonsils. We have added the definition of infection and carriage in the revised manuscript.

2. From the sequence data, it could be informative to also perform analysis of mobile elements such as phage, ICE, etc.

>>**Response:** We performed an association on the pan-genome with infection, sore throat, and antibiotic use, which did not show a significant association. Since mobile elements are present and annotated in the pan-genome, we do not think they deserve a specific subanalysis as we did for the virulence genes. We have added the plasmid information in Supplementary Table S1.

3. Line 84 - what is meant by a smear infection?

>>**Response:** by smear infection, we meant "acquired via direct contact infections or via droplets". Thank you for this comment. We have made the changes to avoid misunderstandings.

4. Line 148 - how was the threshold of 18 SNPs determined?

>>**Response:** The threshold was based on the SNP distribution in the Minimum spanning tree which showed a gap between 18 SNPs and the next SNP distance (39 SNP) as depicted in figure 2. We have now added the value in the branches of the figure 2. As this definition of the threshold is not published, we compared our results to Average Nucleotide Identity (ANI) and our cluster showed good convergence with the clustering based on 99.99% ANI as described by Rodriguez et al. 2023. Only two isolate differs in there clustering, GAS-G0867T would be considered in the cluster13 with ANI but the SNPs distance was 39 SNPs and GAS-G0682T would be removed from Cluster06 because the ANI was 99.87 %. Those difference are most likely relying on the fact that the SNP-based threshold is done on the core-genome while the ANI consider the full genomic content (also mobile element). Therefore, we are confident in our clustering based on SNPs and include the comparison to ANI in the supplement.

5. Line 157 - I'm not convinced that transmission equates to virulence. What about disease severity? Related to comment 1, it is not clear if the intrahousehold strains are related to infections or carriage.

>>**Response:** The vast majority (>95%) of intra-household transmission was related to carriage rather than infection. We have therefore removed this sentence, as this is a rather speculative interpretation, and we agree with the reviewer that transmission does not necessarily equate to virulence. There were no associations between disease severity and transmission.

6. An additional limitation to discuss is that the strains were recovered in 2012, so the data may not be representative of currently circulating strains.

>>**Response:** we agree, we have added this to the limitation

Reviewer #2 (Comments for the Author):

The study of Boutin S et al. covers an important topic that is the molecular epidemiology of *S. pyogenes* in low and middle-income countries, partially filling a gap in the knowledge of the diffusion of this pathogen in Sub-Saharan Africa. *Streptococcus pyogenes* is an important health concern, being responsible for over 500,000 deaths per year, and is the principal etiological agent of children pharyngitis with many severe sequelae. However, I am some concerns about it that need to be solved before considering for publication, besides the limitations mentioned by the Authors (small study population and absence of clinical data linked to the isolated strains).

Major issues:

1. Have you found some difference in emm types identified in throat and skin swabs?

>>**Response:** We observed some emm types present only in the skin and throat, but due to the diversity of EMM types and the sample size of the cohort, we did not find a significant statistical association (Fisher test; p -value=0.09648), so we decided not to further explore the relationship between emm type and screening site. We have included these data in the Results section.

2. In Antibiotic resistance and virulence genes you affirm: We do not observe any associations between the AMR prevalence and the usage of antibiotics in the past or previous treatment of infection (Figure 1). But how much is the AMR prevalence and which antibiotics were used? It is difficult extrapolate from the Figure 1 such information.

>>**Results:** We found only 4 AMR genes *lmp*, *tet(L)*, *tet(M)* and *thfT* with prevalences of 100%, 19.7%, 84.21% and 8%, respectively, which were already included in Figure 1. These values have now been added to the text and the data have been added to Supplementary Table S1. Unfortunately, the data on the specific antibiotic used cannot always be reconstructed because many of the study participants could not recall the name of the drug. In the questionnaire, participants could mark "yes" or "no" for each item. Unfortunately, almost none of the participants were able to recall the exact drug they were taking, nor were they able to show the packaging. Therefore, no further information is available on this point.

3. Have you performed antibiotic susceptibilities for each strain?

>>**Response:** Phenotypic AST was performed for all strains for penicillin, erythromycin, clindamycin, azithromycin, tetracycline and cefotaxime using the disk diffusion method. All isolates were beta-lactam susceptible and are therefore not included in the revised Figure 1. For other substances tested, this information is now included in Figure 1 to allow for a better comparison between genotypic and phenotypic resistance.

4. Are they multi drug resistant?

>>**Response:** No, using the definition of multidrug resistance as being resistant to ≥ 3 drug classes, the isolates collected in the study were not multidrug resistant.

5. Have you estimated the possible vaccine coverage with the identified emm types?

>>**Response:** We have now estimated the coverage based on the 30-valent M-protein-based *Streptococcus pyogenes* vaccine on our population. Results have been integrated in the manuscript.

6. Minor issues: To a better understanding of your results, it is advisable to have some tables for antibiotic and virulence genes

>>**Response:** As described in the previous answers, we have now added the presence/absence of the AMR genes and virulence genes in table S1.

Re: Spectrum04265-23R1 (Genomic epidemiology of Streptococcus pyogenes from pharyngeal and skin swabs in Gabon)

Dear Prof. Dennis Nurjadi:

Your manuscript has been accepted, and I am forwarding it to the ASM production staff for publication. Your paper will first be checked to make sure all elements meet the technical requirements. ASM staff will contact you if anything needs to be revised before copyediting and production can begin. Otherwise, you will be notified when your proofs are ready to be viewed.

Sincerely,
Deena Altman
Editor
Microbiology Spectrum

Reviewer #1 (Comments for the Author):

I thank the authors for their thoughtful responses and edits.